# Assessment of the Health Status of Old Trees of *Platycladus orientalis* L. Using UAV Multispectral Imagery

**Daihao Yin** [1,2] , **Yijun Cai** [1,2], **Yajing Li** [1,2], **Wenshan Yuan** [1,2] **and Zhong Zhao** [1,2,*]

[1] College of Forestry, Northwest A&F University, Yangling 712100, China; daihaoy@nwafu.edu.cn (D.Y.); 2021050636@nwafu.edu.cn (Y.C.); lyj123@nwafu.edu.cn (Y.L.); 2020055610@nwafu.edu.cn (W.Y.)

[2] Research Center for the Conservation and Breeding Engineering of Ancient Trees, Yangling 712100, China

[*] Correspondence: zhaozh@nwafu.edu.cn

**Abstract:** Assessing the health status of old trees is crucial for the effective protection and health management of old trees. In this study, we utilized an unmanned aerial vehicle (UAV) equipped with multispectral cameras to capture images for the rapid assessment of the health status of old trees. All trees were classified according to health status into three classes: healthy, declining, and severe declining trees, based on the above-ground parts of the trees. Two traditional machine learning algorithms, Support Vector Machines (SVM) and Random Forest (RF), were employed to assess their health status. Both algorithms incorporated selected variables, as well as additional variables (aspect and canopy area). The results indicated that the inclusion of these additional variables improved the overall accuracy of the models by 8.3% to 13.9%, with kappa values ranging from 0.166 and 0.233. Among the models tested, the A-RF model (RF with aspect and canopy area variables) demonstrated the highest overall accuracy (75%) and kappa (0.571), making it the optimal choice for assessing the health condition of old trees. Overall, this research presents a novel and cost-effective approach to assessing the health status of old trees.

**Keywords:** health status; multispectral imagery; old trees of *Platycladus orientalis* L.; machine learning algorithms

## 1. Introduction

Old trees hold great significance as living cultural relics, serving as invaluable historical, cultural, scientific, and ecological resources. They are keystone structures in forests, woodlands, savannas, agricultural landscapes, and urban areas, playing unique ecological roles not provided by younger, smaller trees [1]. Furthermore, they have an important place in the human psyche and have many human cultural and aesthetic values [2]. Within ancient tree communities, multiple single or diverse tree species grow together in a relatively concentrated manner, highlighting their exceptional significance and value. However, the survival of old trees is constantly threatened by various abiotic and biotic factors. Hence, it becomes essential to ensure the accurate and timely monitoring of stand and tree health status for the effective health management of old trees [3,4]. Traditionally, the assessment of the health status of trees relies on time-consuming field sampling and observations of symptoms exhibited on the trunks and foliage of trees, with a high degree of uncertainty, only feasible at the plot-scale [5]. Remote sensing enables the acquisition of forest health indicators based on spectral or structural features derived from sensor data in an objective, quantitative, and repetitive manner at multiple spatial scales [6]. In forests of old trees, up-to-date and regularly acquired information becomes a key requirement. Besides timeliness, a very high spatial resolution is also critical in sustainable forest management [7].

Unmanned aerial vehicles (UAVs) offer a promising solution to this issue, as they enable high-intensity data collection at a lower operational cost, providing a more frequent and comprehensive method for assessing the health of old trees [8]. Minařík and

Langhammer [9] used a UAV equipped with a multispectral sensor to distinguish the boundary categories represented by healthy and dead trees, which presents a new methodological approach for the assessment of spatial and qualitative aspects of forest health. Nguyen et al. [10] used UAV images and deep learning to identify individual sick fir trees (*Abies marriesii*) in insect-infested forests and managed to correctly detect/classify 78.59% of all tree classes (39.64% for sick fir).

From the literature survey, in the field of forest health assessment based on UAVs, the remaining works have mainly focused on detecting forest damage [11,12] rather than individual trees, specifically old trees. Most are solely based on RGB sensors [13,14] and limited to spectral and texture information [15,16]. Studies have been conducted on low tree density and low levels of structural complexity, but forest conditions may pose unique challenges for detection [16]. In forests, many large branches overlap, which affects the segmentation of individual trees, and some studies have errors in the acquisition of spectral features due to the angle of elongation. With respect to the aforementioned studies and their limitations, we propose a more diverse and detailed methodological framework, using a combination of spectral and textural information and two algorithms (support vector machines (SVM) and random forest (RF)), followed by introducing additional variables to improve performance and mapping.

Machine learning algorithms are widely used to classify the health of individual trees [5]. Random forest (RF) and support vector machine (SVM) are representative [17]. RF is a machine learning method with a large data processing capacity, fast operation speed, high noise immunity, and an ability to suppress overfitting, based on the generation of classification trees and on the aggregation of their results [18]. SVM is one of the classic machine learning techniques that learns by example to assign labels to objects based on statistical learning theory [19]. SVM is preferred for its ability to perform better with limited training samples. It utilizes only the subset of the training samples that define the location of the SVM optimum hyperplane.

Large old trees are among the most imperiled organisms on earth, and their protection demands innovative approaches to management and monitoring over unprecedented time frames [2]. This study provides useful information for assessing–mapping the health status of old trees aiming to support precision health management and decision making for individual old trees. Here, based on multispectral data collected by a small UAV, our research aims to (1) verify the performance of the rapid health assessment of old trees framework based on UAVs; (2) compare the advantages and disadvantages of different algorithms.

## 2. Material and Methods

### 2.1. Study Area

The central part of the Huangling National Forest Park (Figure 1), located in the southern area of Yan'an City, Shaanxi Province, Northwestern China (35°35′33″–35°35′14″ N, 109°15′29″–109°15′50″ E), was selected as the test area. There are more than 200 centenarian trees here. The total area of the study area is 130,000 m$^2$, and the altitude is 940–965 m. It belongs to the temperate continental monsoon climate, with an average annual temperature of 9.4 °C and an average annual rainfall of 568.8 mm. In the study area, the high-density forest is mainly composed of *Platycladus orientalis*.

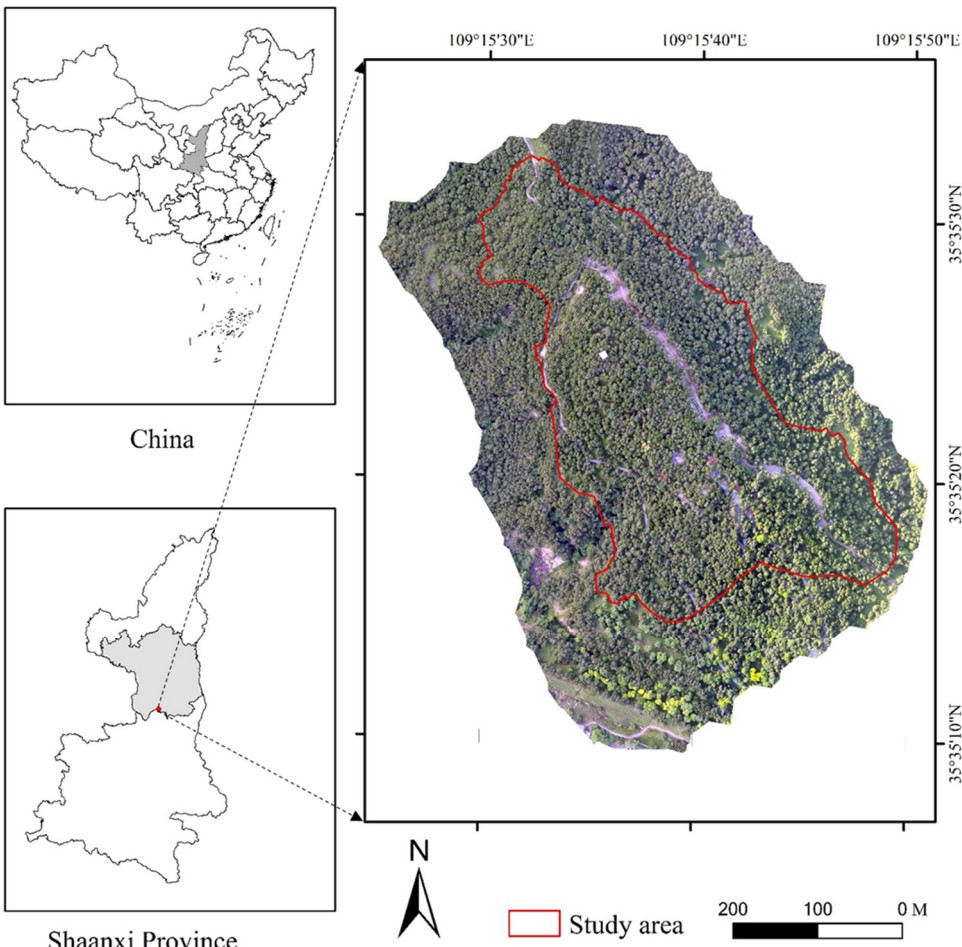

**Figure 1.** Location of the study area.

*2.2. Field Measurement Data*

Field-based sampling was carried out to provide reference data for each forest health status. In the field, 142 old trees were positionally located using a high-precision South RTK GPS (Yinhe Plus 1), and their heights, diameters at breast height, sites and crown widths in the east–west and north–south directions were recorded. The field survey was conducted in July 2022. To assess the health of each sampled tree, twelve parameters were used to evaluate crown vigor and degradation [20]. Each parameter had five-level evaluation criteria, ranging from 0 (best) to 4 (worst), and the evaluation criteria for each level had well-defined visual characteristics of trees. The standardized data of the 12 evaluation indices were subjected to the Kaiser–Meyer–Olkin (KMO) test and the Bartlett spherical test to determine the suitability of principal component analysis. The common factor variance of each index was calculated by principal component analysis, and the proportion of the variance of each common factor to the sum of the common factor variance was further calculated, which was used as the weight of the assessment index. The overall health score of each tree was then obtained. The K-means clustering method was used to evaluate the health status of the old trees, and the ancient trees were divided into three categories: healthy, declining, and severe declining (Table 1). The field data were used as a reference for modelling the predicted tree health status. In the above process, Excel was used for statistical tabulation and IBM SPSS Statistics 27 for data analysis and processing.

**Table 1.** Tree health condition evaluation according to above-ground parts and count of old trees.

| Assessment Items | Evaluation Benchmark | | | | | Score | Weight (%) |
|---|---|---|---|---|---|---|---|
| | 0 | 1 | 2 | 3 | 4 | | |
| Tree vigor | Vigorous growth | Adversely affected | Apparent weakness | Extremely poor | Almost dead | | 9.62 |
| Tree form | Natural tree form | Nearly natural tree form but some exceptions | Natural tree form partially damaged | Natural tree form damaged and deformed | Natural tree form damaged completely | | 10.31 |
| Branch access | Normal | Having a certain but not obvious influence | Shorter and thinner branches | Branches extremely shortened, internodes inflated | Only having lower growth branches | | 9.76 |
| Upper branches and tree apex mortality | None | Not obvious | Many | A great many | No tree apex and branches | | 9.08 |
| Lower branches mortality | None | Not obvious | Some and some broken | Many, mostly broken | Without healthy branches | | 7.96 |
| Damage of trunk and large branches | None | Rarely and having been restored | Obvious | Obvious and broken | Defect in the upper part | | 7.26 |
| Foliage density | Branch and leaf density equilibrium | Normal foliage density | Relatively sparse | Few branches, sparse | Dead branches | | 9.07 |
| Size of leaf buds | Leaf (bud) is sufficiently large | Some leaves (bud) smaller | Most buds smaller | All significantly smaller | Only a small number of buds present and smaller | | 8.89 |
| Foliage colors | Almost thick green | Green | Some obvious yellow/brown leaves | Almost light green | All yellow/brown leaves | | 9.11 |
| Bark damage (peeled/necrosis) | No damage | Few holes, no significant damage | Old scars | Wound decayed significantly | Large hole or severe peeling | | 5.14 |
| Bark metabolism | Fresh bark, strong metabolism | Most of the bark fresh, few locations with weak individual metabolism | Apparent lack of vigor, weak metabolism | Almost all bark without vigor | Most of the bark necrotic | | 5.60 |
| Germination and sprouting | Large amount of foliage, much germination and sprouting | Large amount of foliage, some green shoots sprouting | Less foliage, fewer green shoots sprouting | Little foliage, few green shoots sprouting | No germination and sprouting | | 8.21 |
| Degree of senescence = the sum of the products of the indicator scores and their weights | | | | | | | Final score |
| Final score | <1.40 | 1.40–1.67 | 1.67–2.20 | 2.20–2.48 | >2.48 | | |
| Grade | I | II | III | IV | V | | |
| Senescent degree | Healthy | | Declining | | Severe declining | | |
| Count | 53 | | 68 | | 21 | | |

### 2.3. UAV-Based Multispectral Data

The multispectral imagery was collected by a multispectral UAV (DJI Phantom 4, Shenzhen, China) equipped with a real-time kinematic (RTK) module. The integrated multispectral imaging system includes one visible light (RGB) camera and five multispectral cameras, blue light (B), green light (G), red light (R), red edge, and near infrared (NIR), responsible for visible light imaging and multispectral imaging, respectively.

The multispectral data were acquired from 10:00 to 11:30 on 22–23 July 2022 in the study area. The weather was clear, and the surface temperature was about 25–30 °C. The flight height was 80 m, and the front and side overlap were set to 80% and 75%, respectively.

When processing the UAV data, we transformed the projection coordinate system to WGS 1984 UTM Zone 49 N. The digital surface model (DSM) and orthophotos of the study area were generated by Terra v2.3.3. In this study, the canopy height model (CHM), subtracting the digital elevation model (DEM) from the digital surface model (DSM), as extracted from the DSM using tools in ArcMap 10.8 [21]. Compared with object-oriented image segmentation, object segmentation based stereoscopic information is more conducive to the recognition of single tree structural features, and the marker-controlled watershed algorithm (MCWS) was used for individual tree segments [22]. We depicted the crown shape of 142 old trees as a reference value to verify the results of single tree segmentation (Figure 2a). Then, the segmentation states were divided into true positive (TP), false positive (FP), false negative (FN). Two indexes were used to evaluate the accuracy of results: overall accuracy and F-score. Finally, we obtained a high accuracy (0.836) and an available F-score (0.882). The individual tree segmentation was based on R Studio 4.2 (Figure 2b).

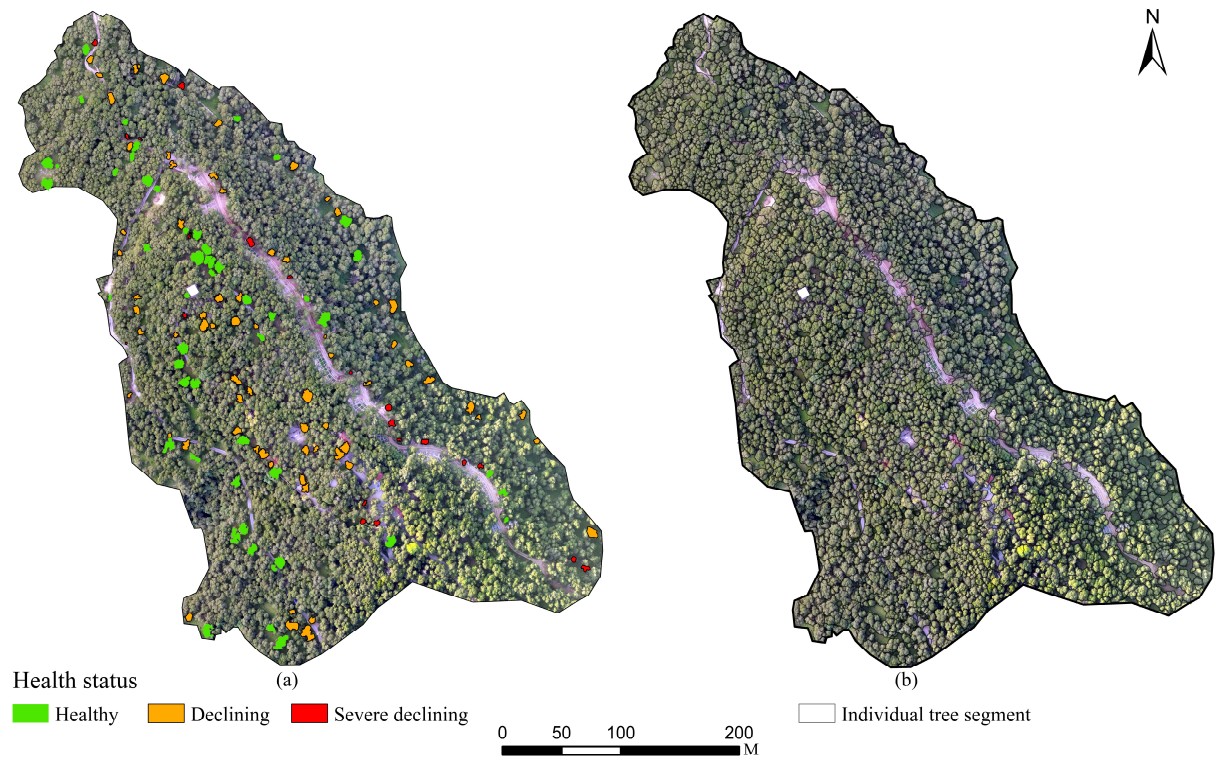

**Figure 2.** Health status of old trees and individual tree segments. (**a**) The distribution of health status of old trees; (**b**) individual tree segments.

*2.4. Feature Extraction*

We extracted candidate explanatory variables from the UAV data at the canopy scale, including both spectral and 3D structural features of the tree canopy.

We used the set of 31 selected vegetation indices to extract canopy values, and we used eight grey level co-occurrence matrices (GLCMs) to extract the texture variables NDVI and DSM, which are useful for describing canopy structure (Table 2) [23]. Mean values were then calculated using all pixels within the canopy. All of the texture variables were calculated using a window size of 3 × 3 pixels and a shift of 45 degrees. All of the above data processing and analysis steps were carried out using the ArcMap 10.8 software and the ENVI 5.4 software.

**Table 2.** Vegetation indices and texture variables with their corresponding formula and reference.

| Class | Variable | Formula | Reference |
|---|---|---|---|
| Vegetation indices | NDVI | $(Nir - R)/(Nir + R)$ | [24] |
| | NDWI | $(Rededge - Nir)/(Rededge + Nir)$ | [25] |
| | RG | $R/G$ | [26] |
| | GB | $G/B$ | [26] |
| | DVI | $Nir-R$ | [27] |
| | MSAVI | $0.5[(2Nir + 1) - \sqrt{(2Nir + 1)2 - 8(Nir - R)}]$ | [28] |
| | MSR | $(Nir/R - 1)/\sqrt{Nir/R + 1}$ | [29] |
| | NDGI | $(G - R)/(G + R)$ | [30] |
| | RVI | $Nir/R$ | [31] |
| | SAVI | $1.5(Nir - R)/(Nir + R + 0.5)$ | [32] |
| | WDEVI | $(0.1Nir - R)/(0.1Nir + R)$ | [33] |
| | ARVI | $(Nir - 2R + B)/(Nir + 2R - B)$ | [34] |
| | ARVI2 | $-0.18 + 0.17(Nir - R)/(Nir + R)$ | [34] |
| | WBRVI | $(0.2Nir - R)/(0.2Nir + R)$ | [33] |
| | CVI | $Nir \times R/G2$ | [35] |
| | GDVI | $Nir - G$ | [36] |
| | EVI | $2.5(Nir - R)/(Nir + 6R - 7.5B + 1)$ | [37] |
| | EVI2 | $2.4(Nir - R)/(Nir + R + 1)$ | [38] |
| | EVI2-2 | $2.5(Nir - R)/(Nir + 2.4R + 1)$ | [39] |
| | GARI | $[Nir - (G - (B - R))]/[Nir - (G + (B - R))]$ | [40] |
| | GBNDVI | $(Nir - (G + B))/(Nir + (G + B))$ | [41] |
| | GRNDVI | $(Nir - (G + R))/(Nir + (G + R))$ | [41] |
| | MRVI | $(RVI - 1)/(RVI + 1)$ | [42] |
| | ANDVI | $(0.5Nir - R)/(0.5Nir + R)$ | [43] |
| | RDNDVI | $(Rededge - R)/(Rededge + R)$ | [44] |
| | PNDVI | $(Nir - (R + G + B))/(Nir + (R + G + B))$ | [41] |
| | RBNDVI | $(Nir - (R+B))/(Nir + (R + B))$ | [41] |
| | LCI | $(Nir - Rededge)/(Nir + R)$ | [45] |
| | NDRE | $(Nir - Rededge)/(Nir + Rededge)$ | [46] |
| | OSAVI | $(Nir - R)/(Nir + R + 0.16)$ | [47] |
| | GNDVI | $(Nir - G)/(Nir + G)$ | [40] |
| Texture | DSM_GLCM and NDVI_GLCM(window size of $3 \times 3$ pixels and a 45 degree shift) | "mean", "variance", "homogeneity", "contrast", "dissimilarity", "entropy", "second_ moment", "correlation" | [23] |

Note: R, G, B, Nir and Rededge represent spectral reflectance of red, green, blue, near-infrared and red edge bands, respectively.

The variables were further analyzed using the Boruta feature selection algorithm [48], which can reduce the influence of a large number of explanatory variables. The Boruta algorithm consists of the following steps [48]: 1. Extend the information system by adding copies of all variables (the information system is always extended by at least five shadow attributes, even if the number of attributes in the original set is lower than 5); 2. Shuffle the added attributes to remove their correlations with the response; 3. Run a random forest classifier on the extended information system and gather the Z scores computed; 4. Find the maximum Z score among shadow attributes (MZSA), and then assign a hit to every attribute that scored better than MZSA; 5. For each attribute with undetermined importance perform a two-sided test of equality with the MZSA; 6. Deem the attributes which have importance significantly lower than MZSA 'unimportant' and permanently remove them from the information system; 7. Deem the attributes which have importance significantly higher than MZSA 'important'; 8. Remove all shadow attributes; 9. Repeat the procedure until importance is assigned for all the attributes or the algorithm has reached the previously set limit of the random forest runs. We introduced aspect and crown area as additional variables. The aspect was divided into four types (Figure 3.) based on DEM data from DSM in individual tree segmentation [49], and we set two data sets (selected variables, selected and additional variables) to build RF and SVM models separately.

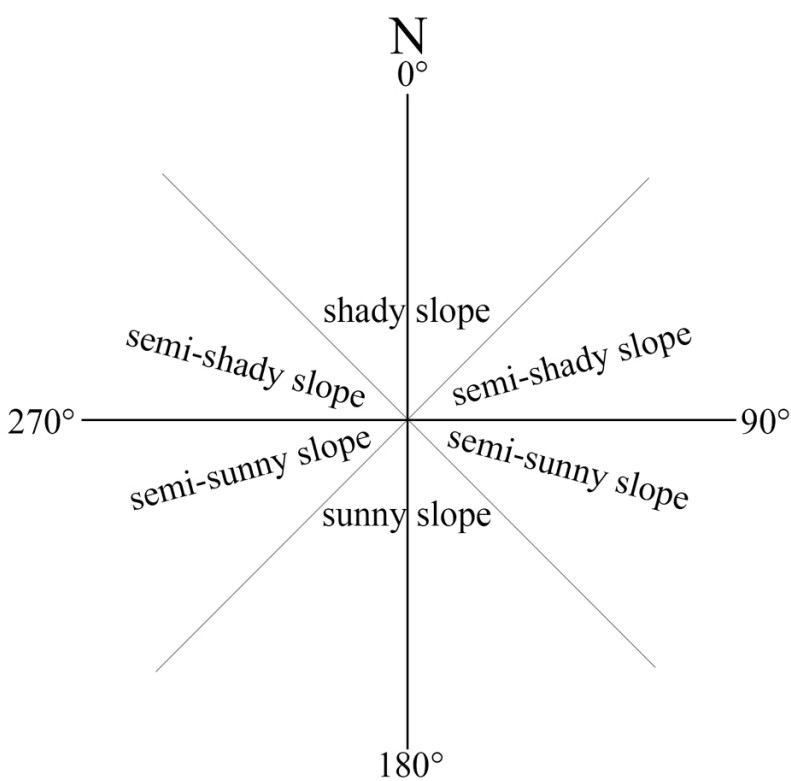

**Figure 3.** Schematic diagram of aspect division [43].

Training/validation data extraction at the crown level was based on the identification and manual delineation on the RGB orthomosaics of the 142 old trees surveyed in the field and stored as polygon vectors. Pixel values of the candidate explanatory variables were retrieved for each tree crown polygon using the extraction function embedded in ArcMap 10.8. The extracted values were used to calculate the mean of the variables for each individual canopy. The derived statistics were stored in a database and used as the input data for further analysis using RF and SVM classifiers.

### 2.5. Tree Health Status Modelling and Mapping of P. orientalis Health Status

SVM and RF were used to assess the health status of old trees. The two models were implemented with the "e1071", and "random Forest" packages in R, version 4.2.1, respectively. During model building, the field measurement data were randomly divided into a training set (75%) and a validation set (25%). The training set was used to build scoring models, and the validation set was used to validate the models. Confusion matrices were constructed using the validation samples and corresponding classification results from the four scenarios described in Section 2.4. A confusion matrix is a table layout that allows a visualization of the performance of a supervised learning algorithm. Each column of the matrix represents the instances in a predicted class, while each row represents the instances in an actual class. Producer's accuracy (PA), user's accuracy (UA), overall accuracy (OA), and kappa coefficients were calculated and used to assess accuracy. Producer accuracy is the probability that ground truth reference data for the category is correctly classified. User accuracy is the probability that a target is correctly classified in a category. Overall precision is the number of all correct classifications as a percentage of the total number of extractions. The kappa coefficient is a metric used to test consistency. By comparing the classification accuracies of the different scenarios, the optimal scenario of the feature combination and classification algorithm was determined and then used to predict the health status of old trees in the whole area.

## 3. Result

### 3.1. Field Health Assessment

We used principal component analysis (PCA) to obtain the overall health score of old trees, and we obtained KMO = 0.878 > 0.5, Bartlett's test of sphericity $p < 0.001$, representing a suitable candidate for principal component analysis. Based on the criterion of eigenvalue greater than 1, three common factors were extracted using principal component analysis and using the maximum variance method; the eigenvalue of component 1 was 6.107, the eigenvalue of component 2 was 1.234, and the eigenvalue of component 3 was 1.034, and the variance contribution ratio of these three was 69.80% (Table 3). The weights of the components and the overall health score were obtained by calculation (Table 2). Then, after K-means clustering, they were divided into three categories, healthy, declining, and severe declining, and the numbers were 53, 68, and 21, respectively.

**Table 3.** Total Variance Explained.

| Component | Initial Eigenvalue | | | Extract the Sum of the Squares of the Loads | | | Rotational Load Sum of Squares | | |
|---|---|---|---|---|---|---|---|---|---|
| | Total | Percentage of Variance | Accumulated % | Total | Percentage of Variance | Accumulated % | Total | Percentage of Variance | Accumulated % |
| 1 | 6.107 | 50.894 | 50.894 | 6.107 | 50.894 | 50.894 | 3.203 | 26.694 | 26.694 |
| 2 | 1.234 | 10.283 | 61.177 | 1.234 | 10.283 | 61.177 | 3.038 | 25.316 | 52.011 |
| 3 | 1.034 | 8.62 | 69.797 | 1.034 | 8.62 | 69.797 | 2.134 | 17.787 | 69.797 |
| 4 | 0.748 | 6.231 | 76.029 | | | | | | |
| 5 | 0.671 | 5.595 | 81.623 | | | | | | |
| 6 | 0.592 | 4.937 | 86.56 | | | | | | |
| 7 | 0.439 | 3.659 | 90.219 | | | | | | |
| 8 | 0.344 | 2.865 | 93.083 | | | | | | |
| 9 | 0.271 | 2.26 | 95.344 | | | | | | |
| 10 | 0.223 | 1.862 | 97.206 | | | | | | |
| 11 | 0.218 | 1.816 | 99.022 | | | | | | |
| 12 | 0.117 | 0.978 | 100 | | | | | | |

### 3.2. Feature Selection

In the feature selection, the Z score of the most important shadow attribute clearly separates important and unimportant attributes. Red, yellow, and green boxplots represent Z scores of, respectively, rejected, tentative, and confirmed attributes. Four variables were selected, including three vegetation indices (NDVIredge, NDGI, and rbNDVI) and one texture measure (DSM-mean) (Figure 4).

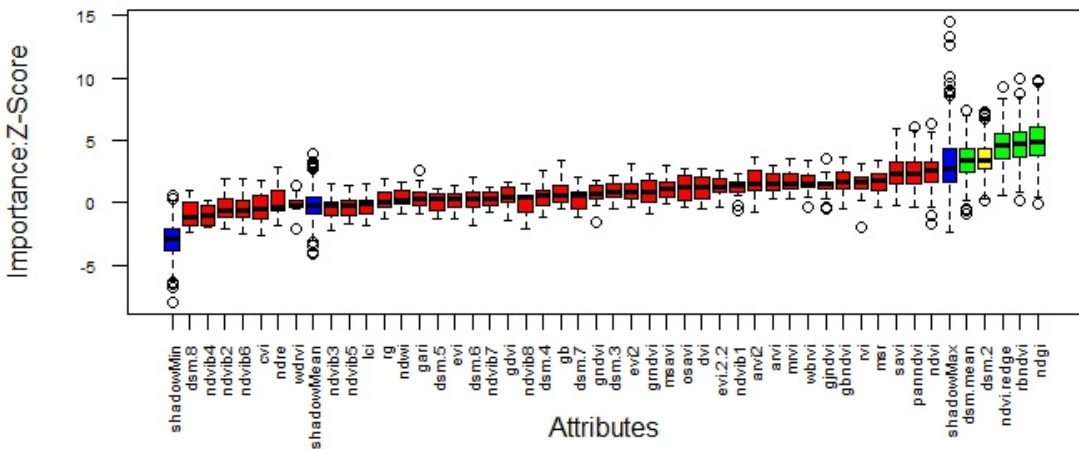

**Figure 4.** Boruta result plot for data. Blue boxplots correspond to the minimal, average, and maximum Z score of a shadow attribute. Red, yellow, and green boxplots represent Z scores of, respectively, rejected, tentative, and confirmed attributes.

### 3.3. Model Comparison

SVM performed better than RF with the same selected features in terms of overall accuracy and kappa coefficient. As shown in Table 4, the overall accuracies and kappa values based on selected features were 55.6% and 0.197 for SVM and 61.1% and 0.338 for RF. In SVM, the main errors came from a misclassification of the healthy and severe declining as declining, resulting in low PA for the healthy (0.14) and severe declining (0.4). In RF, the main error came from misclassifying severe declining as declining, resulting in a low PA for severe declining (0.2).

**Table 4.** Accuracies of old tree health conditions based on selected features with Support Vector Machine (SVM) and Random Forest (RF) algorithms.

| Data and Method | Classified Levels | Reference Data | | | Total | UA |
| --- | --- | --- | --- | --- | --- | --- |
| | | **Healthy** | **Declining** | **Severe Declining** | | |
| | Healthy | 2 | 1 | 0 | 3 | 0.67 |
| | Declining | 12 | 16 | 3 | 31 | 0.52 |
| Selected with SVM | Severe declining | 0 | 0 | 2 | 2 | 1 |
| | Total | 14 | 17 | 5 | 36 | |
| | PA | 0.14 | 0.94 | 0.40 | | |
| | OA | | 55.6% | | Kappa | 0.197 |
| | Healthy | 11 | 6 | 1 | 18 | 0.61 |
| | Declining | 3 | 10 | 3 | 16 | 0.62 |
| Selected with RF | Severe declining | 0 | 1 | 1 | 2 | 0.50 |
| | Total | 14 | 17 | 5 | 36 | |
| | PA | 0.78 | 0.59 | 0.2 | | |
| | OA | | 61.1% | | Kappa | 0.338 |

### 3.4. Models with Crown Area and Aspect Variables

For the models with crown area and aspect variables, accuracy was improved in terms of both overall accuracy and kappa (Table 5). The overall accuracy of A-SVM and A-RF was 63.9% and 75%, respectively, showing an increase of 8.3% and 13.9%. The kappa coefficients of A-SVM and A-RF were 0.363 and 0.571, representing an increase of 0.166 and 0.233, respectively. After adding variables, all categories performed better, including UA and PA. The A-RF model, which had the higher OA (75%) and kappa (0.571), was considered optimal for estimating the health status of old trees.

**Table 5.** Accuracies of old tree health conditions based on selected and added features with Support Vector Machine (SVM) and Random Forest (RF) algorithms.

| Data and Method | Classified Levels | Reference Data | | | Total | UA |
| --- | --- | --- | --- | --- | --- | --- |
| | | **Healthy** | **Declining** | **Severe Declining** | | |
| | Healthy | 4 | 1 | 0 | 5 | 0.8 |
| | Declining | 10 | 16 | 2 | 28 | 0.57 |
| Selected & Area & Asp with SVM(A-SVM) | Severe declining | 0 | 0 | 3 | 3 | 1.0 |
| | Total | 14 | 17 | 5 | 36 | |
| | PA | 0.28 | 0.94 | 0.60 | | |
| | OA | | 63.9% | | Kappa | 0.363 |
| | Healthy | 12 | 3 | 1 | 16 | 0.75 |
| | Declining | 1 | 14 | 3 | 18 | 0.78 |
| Selected & Area & Asp with RF(A-RF) | Severe declining | 1 | 0 | 1 | 2 | 0.50 |
| | Total | 14 | 17 | 5 | 36 | |
| | PA | 0.86 | 0.82 | 0.2 | | |
| | OA | | 75% | | Kappa | 0.571 |

*3.5. Spatial Distribution of Old Trees with Different Health Conditions*

The A-RF model, based on a combination of crown area and aspect characteristics, was applied to the whole study area and produced a spatial distribution of the health status of old *P. orientalis* trees (Figure 5). The number and proportion of old trees in each health category was summarized in Table 6. About two-fifths of old trees were healthy, and more than half were in decline. It is necessary to take urgent measures for the declining old trees.

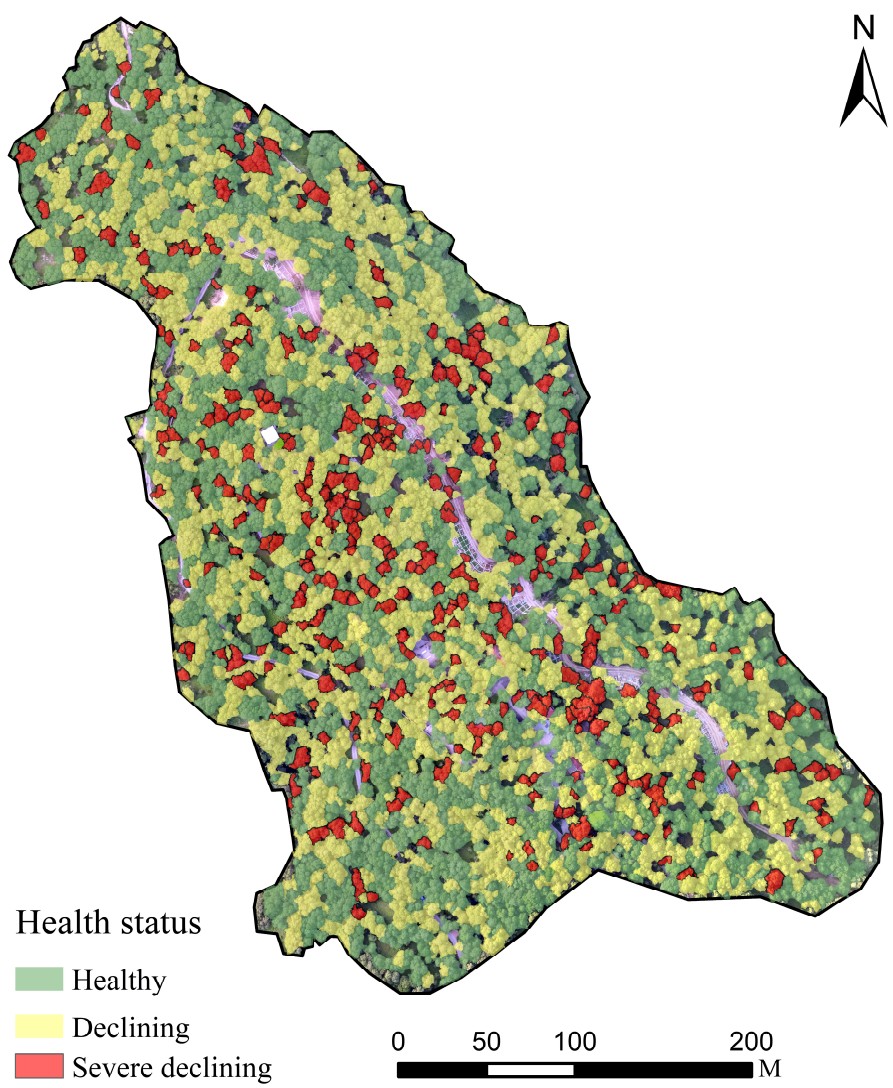

**Figure 5.** Spatial distribution of old trees with different health status.

**Table 6.** The number and proportion of old trees at different health conditions.

| Health Status | Number of Trees | Percentage of Study Area (%) |
|---|---|---|
| Healthy | 1125 | 43.4 |
| Declining | 1121 | 43.2 |
| Severe declining | 349 | 13.4 |
| Total | 2595 | 100 |

## 4. Discussion

To our knowledge, this is the first study to assess the tree-level health status of old trees using UAVs; although there are limitations, our study may lead to improving the health management of old trees.

### 4.1. Health Status and Management of Old Tree

In our study, the number of healthy and declining trees was more than 40%, and the number of severe declining trees was 349, over 10%. Although incorporated into the park management system, it was open to tourists. Habitat alteration associated with park establishment activities (e.g., road access and soil compaction) and increased numbers of visitors may have a negative impact on the growth of trees in these sites [50]. Old trees can be susceptible to disease, insect attack, and dieback [51,52] and are at risk of decline [53]. Therefore, more than half were in the declining and severe declining categories. For trees in decline, we should take timely mitigation measures according to the actual situation. Introducing symbiotic organisms, both below ground and in the above-ground forest microbiome, has the potential to facilitate plant growth [54]. The system developed by Osnabriick for the further protection of crowns threatened by possible failure was a useful path to reduce the breakage of crown parts and large branches [55]. Some large cavities should be filled with special materials, such as zinc- and copper-based nanocompounds [56], to prevent further damage.

### 4.2. Selecting Variables

Vegetation Indices and texture measures derived from multispectral UAV imagery are widely used for assessing health status. In our study, we chose 47 variables, including 31 vegetation indices and 16 texture measures. In order to remove redundant features as well as reduce overfitting, we used the Boruta feature selection algorithm to select variables. Three vegetation indices (NDVIredge, NDGI, and rbNDVI) and one texture measure (DSM-mean) were selected. The spectra of problematic canopies will differ significantly from the reflectance spectra of normal canopies [16]. NDVI redge and rbNDVI are considered to be a good vigor indicator [57]. NDGI is an indicator of changes in the status quo of the vegetation [58]. They have been widely used to characterize canopy status [59,60]. Texture is a tool for documenting stand structure. We extracted texture features from DSM and NDVI; only DSM-mean was selected, and the results were different from the study by Guerra-Hernández, J. et al. [23]. The window size as well as the orientation of the texture features have a significant effect on the results [61]. Further tests should therefore be carried out.

### 4.3. Performance of Two Models

In this study, SVM and RF were used to estimate the health status of old trees of *P. orientalis* by combining texture variables and vegetation indices based on UAV images. Compared to SVM, RF had performed better in terms of both accuracy (61.1%) and kappa values (0.338). When performing multi-classification, RF obtained better classification results [62,63]. RF outperformed SVM in their ability to generalize and handle multi-dimensional data. The training sample set affects the performance of both classifiers SVM and RF [64,65]. RF gave better results when there was a large number of training samples available [15]. Our training samples share of 4% was much larger than 0.25% [62], although SVM had good performance on imbalanced training datasets [16]. However, due to variations in stem density and spacing and the range of tree heights and sizes, there is tremendous variability and overlap in the spectral signature of trees in old-growth *P. orientalis* stands [66,67], which make the spectrum unusual and low in accuracy. In addition, our health status results were derived from a qualitative analysis of old-growth trees rather than the more traditional discoloration of the canopy [68,69]. Further research needs to be used to improve accuracy.

### 4.4. Adjusting Models

In the field study, we found that there were significant differences ($p < 0.05$) in the old-growth forests in terms of aspect and crown areas. Aspect, an important topographic factor, can often create a local microclimate by altering ecological factors such as light, temperature, water, and soil, which affects the health of the community [70]. The canopy is where the tree

photosynthesizes, and its area reflects its health in old trees. Therefore, we included aspect and area variables of canopy that influence health status in our estimation models. For models with additional variables, the increase in overall accuracy and kappa value varied from 8.3% to 13.9% and from 0.166 to 0.233, respectively. A-RF performed better in OA (75%) and kappa (0.571), indicating a great potential for incorporating aspect and crown area into old tree health assessment. In the A-RF model, healthy and declining categories were identified with high UA and PA (over 70%). In particular, the PA of the healthy and declining categories were 0.86 and 0.82, showing our results could be used to guide practical forest management. A-RF had low pairwise correlations, and classification accuracy could be significantly improved by aggregating the results of many classifiers that have little bias by averaging or voting [71]. However, the UA and PA of the severe declining category were 0.5 and 0.2, respectively, the reason being that the number of severe declining trees in the study area was small, so there was a high probability of complete misclassification or correct classification [22]. Roope Näsi et al. [72] used hyperspectral imagery to identify mature Norway spruce (*Picea abies* L. Karst.) trees suffering from infestation, and the best results for the overall accuracy were 76% (Cohen's kappa 0.60) when using three color classes (healthy, infested, dead). Our study obtained similar accuracy through more low-cost equipment.

Old-growth stands are less structurally diverse [73]. The development of old-growth attributes is modified by site position [74], and characteristics such as the percentage of broken-topped crowns increase with stand age [73]. Therefore, the introduction of aspect, a major site factor influencing the growth and development of a single plant, and canopy area, a direct reflection of the growth status of individual trees, into the health assessment model reflects the health of old trees and significantly improves the estimation accuracy. Studies on the integration of CHM feature [75], normalized digital surface models [10], and canopy cover [76] into the model also proved that the introduction of factors reflecting the growth status can improve the accuracy of health estimation. In the future, hyperspectral and LiDAR data should be obtained and used to assess the health status of old trees of *P. orientalis* L.

## 5. Conclusions

In this study, we used two machine learning methods, SVM and RF, to assess the health status of old trees, based on UAV multispectral data. Two models obtained a general performance with traditional methods (only vegetation indices and textures variables), but RF performed better, especially in the detection of declining trees. Furthermore, we introduced two additional variables (aspect and canopy area) in the health assessment models, obtaining a significantly improved performance both in accuracy and kappa value. In the assessment of health status, A-RF (RF with aspect and canopy area variables) with overall accuracy (75%) and kappa (0.571) had a better applicability in the study area. Knowing the health status of old trees is of great importance for forest management. Our study provides a suitable method for assessing the health status of old trees, making it a reality to achieve rapid and reproducible diagnosis which provides a basis for the health management of old trees.

**Author Contributions:** Z.Z. developed and supervised the work. D.Y. investigated field data, analyzed the data and wrote the paper. Y.C., Y.L. and W.Y. contributed to the field investigation and data analysis. All authors have read and agreed to the published version of the manuscript.

**Funding:** This work was financially supported by Multifunctional enhancement of acacia forests in hilly and gully areas and stabilisation and maintenance of vegetation on the Loess Plateau Maintenance Technology (Grant No. 2022YFF1300405).

**Data Availability Statement:** The data presented in this study are available on request from the corresponding author (accurately indicate status).

**Conflicts of Interest:** The authors declare that they have no known competing financial interests or personal relationships that could have appeared to influence the work reported in this paper.

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
