# Peer review of "Assessment of the Health Status of Old Trees of Platycladus orientalis L. Using UAV Multispectral Imagery"

_drones, doi:10.3390/drones8030091_

Round 1

Reviewer 1 Report

Comments and Suggestions for Authors

The article is camera ready. All the best. 

Comments on the Quality of English Language

English is fine. 

Author Response

Dear reviewer,

We feel great thanks for your professional review work on our article.Thank you for your approval.

Reviewer 2 Report

Comments and Suggestions for Authors

This paper needs minor revision. I have indicated corrections to be made consisting of minor changes in the text.

Author Response

Dear reviewer,

We feel great thanks for your professional review work on our article.Thank you for your approval.Those comments are all valuable and very helpful for revising and improving our paper, as well as the important guiding significance to our researches. We have studied comments carefully and have made correction which we hope meet with approval. 

Reviewer 3 Report

Comments and Suggestions for Authors

This study proposes an approach for rapid assessment of health status of old trees. Its main contribution consists in improving the accuracy of two machine learning models by incorporating selected variables and additional variables (aspect and canopy area).

The document is easy to read and follow.

The document is well supported with references although old.

The subject of the paper has great potential of application.

In Table 1 and Table 5 authors present the counting of trees from each class (53 Healthy, 68 Declining and 21 Severe declining). Authors should explain in the document how the class count unbalance was addressed.

In line 135 authors should clarify in the text, in more detail, the variables selection process including the number of variables (4) and the reasoning behind the selection of specifically those variables or reference the section were this is addressed in more detail.

In line 157 authors claim that the dataset was divided into training set (75%) and validation set (25%). Usually, to build a reliable machine learning model, it is required to split the dataset into the training, validation, and test sets. Otherwise, the results might be biased giving the impression of better model performance (accuracy). Please explain and correct in the document.

From Table 4 it is observed that the best overall accuracy obtained with the proposed methodology was 75% which is a very modest result for machine learning approaches. Authors should better and in more detail explain these results in the document.

In line 158 authors refer to the confusion matrices but were not included in the document. Please explain and/or include these as figures.

Comments on the Quality of English Language

The English needs minor spell checking.

Author Response

------------------------------------------------------------------------

Authors response: Dear Reviewer: We greatly appreciate your professional review work on our article. Thank you for your acknowledgement. The comments are valuable and helpful in revising and improving the paper, and they are an important guide for our research. We have carefully studied the comments and made corrections, and we hope that they will be approved.

Comment 1:In Table 1 and Table 5 authors present the counting of trees from each class (53 Healthy, 68 Declining and 21 Severe declining). Authors should explain in the document how the class count unbalance was addressed.

Response 1:The category imbalance problem refers to a highly heterogeneous sample size across categories in a dataset. In our study, the ratio of the number of the declining to the number of the severe declining was 3:1, therefore, in the modelling process, we used stratified sampling to construct the dataset according to the proportion of each category.

Comment 2:In line 135 authors should clarify in the text, in more detail, the variables selection process including the number of variables (4) and the reasoning behind the selection of specifically those variables or reference the section were this is addressed in more detail.

Response 2:Thank you for pointing this out. We agree with this comment. Therefore, I've added the variable selection and description to the document, and you can learn more about it in 153-167 ; 213-219; 273-285.

Comment 3:In line 157 authors claim that the dataset was divided into training set (75%) and validation set (25%). Usually, to build a reliable machine learning model, it is required to split the dataset into the training, validation, and test sets. Otherwise, the results might be biased giving the impression of better model performance (accuracy). Please explain and correct in the document.

Response 3: When constructing a classification model, especially when the sample size is not particularly large, its dataset is usually divided into two parts. For example, in the article "Urban forest monitoring based on multiple features at the single tree scale by UAV ", 143 samples are divided into 105 training samples and 38 validation samples, with a ratio of almost 3:1.

Comment 4:From Table 4 it is observed that the best overall accuracy obtained with the proposed methodology was 75% which is a very modest result for machine learning approaches. Authors should better and in more detail explain these results in the document.

Response 4:After much deliberation, we have further developed these findings, which are set out in papers 312-320.

Comment 5:In line 158 authors refer to the confusion matrices but were not included in the document. Please explain and/or include these as figures.

Response 5:We further elaborate on the content of the confusion matrix and describe the associated accuracy metrics.

------------------------------------------------------------------------

Reviewer 4 Report

Comments and Suggestions for Authors

I think the research results are academically and technically very interesting. However, to increase readers' comprehension, please consider the following points.

1. Introduction:

- Line 25-28: It is necessary to provide a more detailed explanation, along with citations, on why old trees are of significance and value, in a way that is easily understandable and accessible to the readers.

- Line 31-33: The authors argue that the traditional methodology, which involves field sampling and symptom observation, is time-consuming. In section 2.2, the health status of trees is assessed using 12 parameters based on field measurements. However, the results are classified using only three criteria derived from UAV imagery. It is therefore essential to provide a detailed explanation, along with evidence, regarding the significance and value of UAV-based classification in evaluating the health of trees.

- Line 47-48: It is recommended that the authors provide a more detailed explanation on why forest conditions are more challenging, to facilitate easier understanding.

- Line 54: The cited paper ranked the machine learning algorithms as ANN-RF-SVM. Is there a specific reason for not using ANN in this study?

- Line 55: The results seem to be derived from 12th reference, but the lack of a direct citation could lead to misinterpretation as a subjective assertion. Including a citation in the sentence is recommended.

- Line55-62: SVM is explained with a cited paper. However, there is no citation for RF. It is recommended to add a reference for RF.

- Line 63: In the previous paragraph, there was no mention of climate change. Consequently, the assertion in this sentence that it has a negative impact seems to be a logical leap. A comprehensive review and rewrite of the introduction should be needed.

- Line 64: It is unclear what new information is being provided. This concern is related to the comment made on line 31-33. Reviewing prior research suggests that the methodology used is merely an application of existing methods to a different subject. Remote sensing methods can be less accurate than field survey. Without offering any new benefits or value, there seems to be no reason to use remote sensing. Therefore, a clearer elaboration in the introduction on this matter is necessary.

2. Material and Methods

- Section 2.1: If the study aims to assess the health status of trees, it should include an explanation of how the trees in the research area are distributed across different health states.

- Section 2.2: For the paper’s classification into three categories to be credible, it needs to demonstrate that the field measurements data were appropriately acquired, evaluated, and clustered. In other words, without assurance of the reliability of the foundational data, the evaluation of the machine learning model’s performance is meaningless. The results therefore include the field measurement data and analysis.

- Section 2.3: Figure 2 represents the research results. It is important to clearly distinguish between research methods and results. Furthermore, if individuals tree segmentation is emphasized in the introduction, it should be presented in the results. Additionally, there is no explanation for choosing MCWS for individual tree segmentation. Lastly, regarding accuracy analysis, if It’s segmentation, would it be assessed using metrics like intersection of union (IoU)? A detailed explanation of how accuracy was analyzed is necessary.

- Figure 2: 'a' and 'b' are not indicated. The figure needs to be revised.

- Section 2.4: The Boruta feature selection algorithm was used, resulting in a reduction to four variables. There should be an interpretation and discussion on why these were ultimately chosen. Furthermore, such content belongs in the results section.

- Table 2: Shouldn't this apply to vegetation indices as well as GLCM? It might be better to explicitly describe what is presented in Table 2.

- Line 138: Isn't this a discussion of the results? Moreover, what is the basis for this discussion? It is not presented. There is a need to clearly redefine the distinction between the research methods and the results.

- Lines 160-161: It is necessary to explain what each metric is intended to interpret and evaluate.

3. Results

- In the accuracy assessment of the three classification categories, there is a complete lack of discussion on why each category achieves its respective accuracy. Additional explanation is needed. 

Comments on the Quality of English Language

I think the quality of English is generally good.

Author Response

Dear Reviewer:

We greatly appreciate your professional review work on our article. Thank you for your acknowledgement. The comments are valuable and helpful in revising and improving the paper, and they are an important guide for our research. We have carefully studied the comments and made corrections, and we hope that they will be approved.

 Comment 1:- Line 25-28: It is necessary to provide a more detailed explanation, along with citations, on why old trees are of significance and value, in a way that is easily understandable and accessible to the readers.

Response 1:Thank you for pointing this out. I agree with this comment. Therefore, I have  added detailed descriptions and citations related to the significance of ancient trees.

Comment 2: Line 31-33: The authors argue that the traditional methodology, which involves field sampling and symptom observation, is time-consuming. In section 2.2, the health status of trees is assessed using 12 parameters based on field measurements. However, the results are classified using only three criteria derived from UAV imagery. It is therefore essential to provide a detailed explanation, along with evidence, regarding the significance and value of UAV-based classification in evaluating the health of trees.

Response 2:Agree. I have, accordingly, revised UAV-based classification in evaluating the health of trees to emphasize this point.

Comment 3:- Line 47-48: It is recommended that the authors provide a more detailed explanation on why forest conditions are more challenging, to facilitate easier understanding.

Response 3:Agree. I have modified the comment why forest conditions are more challenging.

Comment 4:- Line 54: The cited paper ranked the machine learning algorithms as ANN-RF-SVM. Is there a specific reason for not using ANN in this study?

Response 4:After consulting the relevant literature, we learned that the most commonly used machine learning models in forest health classification are Random Forest and Support Vector Machines, so only these two were chosen for the purpose of the following study.

Comment 5:- Line 55: The results seem to be derived from 12th reference, but the lack of a direct citation could lead to misinterpretation as a subjective assertion. Including a citation in the sentence is recommended.

Response 5:Thank you for pointing this out. I agree with this comment. Therefore, I have  added a reference.

Comment 6:- Line55-62: SVM is explained with a cited paper. However, there is no citation for RF. It is recommended to add a reference for RF.

Response 6:Thank you for pointing this out. I agree with this comment. Therefore, I have  added a  reference for RF.

 Comment 7: Line 63: In the previous paragraph, there was no mention of climate change. Consequently, the assertion in this sentence that it has a negative impact seems to be a logical leap. A comprehensive review and rewrite of the introduction should be needed.

Response 7: Agree. I have revised a comprehensive review and rewrite of the introduction.

Comment 8:Line 64: It is unclear what new information is being provided. This concern is related to the comment made on line 31-33. Reviewing prior research suggests that the methodology used is merely an application of existing methods to a different subject. Remote sensing methods can be less accurate than field survey. Without offering any new benefits or value, there seems to be no reason to use remote sensing. Therefore, a clearer elaboration in the introduction on this matter is necessary

Response 8:Thank you for pointing this out. I agree with this comment. Therefore, I have changed the comment.

Comment 9:Section 2.1: If the study aims to assess the health status of trees, it should include an explanation of how the trees in the research area are distributed across different health states.

Response 9:Agree. I have added a comprehensive discussion for health status and management of old tree.

Comment 10:- Section 2.2: For the paper’s classification into three categories to be credible, it needs to demonstrate that the field measurements data were appropriately acquired, evaluated, and clustered. In other words, without assurance of the reliability of the foundational data, the evaluation of the machine learning model’s performance is meaningless. The results therefore include the field measurement data and analysis.

Response 10:Thank you for pointing this out. I agree with this comment. Therefore, I have added the results for the field measurement data and analysis.

Comment 11:- Section 2.3: Figure 2 represents the research results. It is important to clearly distinguish between research methods and results. Furthermore, if individuals tree segmentation is emphasized in the introduction, it should be presented in the results. Additionally, there is no explanation for choosing MCWS for individual tree segmentation. Lastly, regarding accuracy analysis, if It’s segmentation, would it be assessed using metrics like intersection of union (IoU)? A detailed explanation of how accuracy was analyzed is necessary.

Response 11:In the introduction, individuals tree segmentation is not emphasized,so we don't use loU .We added the reason for choosing MCWS for individual tree segmentation and a detailed explanation of how accuracy.

Comment 12: - Figure 2: 'a' and 'b' are not indicated. The figure needs to be revised.

Response 12:Thank you for pointing this out. I agree with this comment. Therefore, I have advised.

Comment 13:Section 2.4: The Boruta feature selection algorithm was used, resulting in a reduction to four variables. There should be an interpretation and discussion on why these were ultimately chosen. Furthermore, such content belongs in the results section.

Response 13:Thank you for pointing this out. I agree with this comment. Therefore, I have added the feature selection in results.

Comment 14: Table 2: Shouldn't this apply to vegetation indices as well as GLCM? It might be better to explicitly describe what is presented in Table 2.

Response 14:Thank you for pointing this out.I have added descriptions in Table 2.

Comment 15:- Line 138: Isn't this a discussion of the results? Moreover, what is the basis for this discussion? It is not presented. There is a need to clearly redefine the distinction between the research methods and the results.

Response 15:Thank you for pointing this out.I have redefined the distinction between the research methods and the results and modified the discussion.

Comment 16: - Lines 160-161: It is necessary to explain what each metric is intended to interpret and evaluate.

Resopnse 16:Thank you for pointing this out.I have added the explaination of metrics.

Comment 17:  In the accuracy assessment of the three classification categories, there is a complete lack of discussion on why each category achieves its respective accuracy. Additional explanation is needed.

Resopnse 17:Thank you for pointing this out.I have added the explaination to make it more complete.

Round 2

Reviewer 3 Report

Comments and Suggestions for Authors

Since the main issues pointed out in the previous review were addressed by the authors, I would advise that the manuscript can be accepted for publication in the present state.

Best regards!

Reviewer 4 Report

Comments and Suggestions for Authors

The manuscript has been sufficiently revised, taking into account the opinions of the reviewer.